# Predicting successful draft outcome in Australian Rules football: Model sensitivity is superior in neural networks when compared to logistic regression

Jacob Jennings[1,2☯], Jay C. Perrett[3‡], Daniel W. Wundersitz[1‡], Courtney J. Sullivan[1‡], Stephen D. Cousins[1‡], Michael I. Kingsley[1,4☯]*

1 Holsworth Research Initiative, La Trobe Rural Health School, La Trobe University, Bendigo, VIC, Australia,
2 La Trobe University Bendigo Pioneers, Bendigo, Australia, 3 PhysiGo Ltd, Wiltshire, England, United
Kingdom, 4 Department of Exercise Sciences, Faculty of Science, University of Auckland, Auckland, New
Zealand

☯ These authors contributed equally to this work.
‡ JCP, DWW, CJS and SDC also contributed equally to this work.
* Michael.Kingsley@Auckland.ac.nz

doi.org/10.1371/journal.pone.0298743

**Editor:** Julio Alejandro Henriques Castro da Costa,
Portugal Football School, Portuguese Football
Federation, PORTUGAL

**Data Availability Statement:** The data supporting
this research are within the manuscript and/or the

## Abstract

Using logistic regression and neural networks, the aim of this study was to compare model
performance when predicting player draft outcome during the 2021 AFL National Draft.
Physical testing, in-game movement and technical involvements were collected from 708
elite-junior Australian Rules football players during consecutive seasons. Predictive models
were generated using data from 465 players (2017 to 2020). Data from 243 players were
then used to prospectively predict the 2021 AFL National Draft. Logistic regression and neu-
ral network models were compared for specificity, sensitivity and accuracy using relative
cut-off thresholds from 5% to 50%. Using factored and unfactored data, and a range of rela-
tive cut-off thresholds, neural networks accounted for 73% of the 40 best performing models
across positional groups and data configurations. Neural networks correctly classified more
drafted players than logistic regression in 88% of cases at draft rate (15%) and convergence
threshold (35%). Using individual variables across thresholds, neural networks (specificity =
79 ± 13%, sensitivity = 61 ± 24%, accuracy = 76 ± 8%) were consistently superior to logistic
regression (specificity = 73 ± 15%, sensitivity = 29 ± 14%, accuracy = 66 ± 11%). Where the
goal is to identify talented players with draft potential, model sensitivity is paramount, and
neural networks were superior to logistic regression.

## Introduction

The identification of characteristics that predict sporting talent is important in recruiting and
developing future elite-level performers [1]. Talent identification is a process that recognises
characteristics in players that are congruent with expert performance, while recruitment is the
process undertaken to acquire the desired talent [1–3]. Performance in team sport begins with

accompanying supporting information file. Additionally, the dataset associated with this study is deposited in the OPAL repository (DOI: https://doi.org/10.26181/24967014).

**Funding:** The author(s) received no specific funding for this work.

**Competing interests:** The authors have declared that no competing interests exist.

the periodic recruitment of identified talent and is then determined by the complex interplay and combination of physical, technical, and tactical elements [4]. Identifying the strengths and weaknesses of individual players that impact draft success can inform recruitment, player development and training decisions [5–7].

Within Australian Rules football (AF), recruitment occurs informally through scouting of the junior participation pathway into the elite-junior talent pathway and formally through a yearly structured draft system in the elite-senior Australian Football League (AFL). Similar to other draft-based sports like Basketball [8] and American Football [9], talented juniors and off-contract players are invited to be part of the AFL's National Draft in November of each year. Therefore, the primary objective of the elite-junior talent pathway is to provide AFL teams with new recruits who possess desirable attributes. Although being drafted does not guarantee a successful career in the AFL, draft success is a critical step in the talent pathway. Consequently, the potential exists to use data from the elite-junior talent pathway to reduce subjectivity in the selection process [5–7].

Predictive modelling has been used extensively in other sports and is now more common both in the literature and its application within AF player recruitment, particularly the draft system [5, 6, 10]. Authors have investigated combinations of physical, in-game movement, and match performance variables that are associated with early ($\leq$5 year) career success and contract renewals within the AFL [11, 12]. Physical testing performance has been used in elite-junior populations to predict selection level or playing status [5–7], and in-game movement and a small number of key technical involvements have been used to predict draft outcome [13, 14]. In a more recent study, authors used anthropometric, physical testing, and in-game movement variables to generate factors and investigate associations with draft success. While results yielded excellent accuracy and specificity, the sensitivity, or the ability to characterise a player with draft potential, was less convincing. Including anthropometric, physical testing, and in-game movement improved the overall accuracy in existing models; however, the relatively poor ability to predict those players who will be drafted limits the application of these models in the recruitment process.

The existing predictive models in AF use traditional statistical techniques like multiple linear regression or logistic regression more suited to linear data and simple relationships, often with poor or unreported sensitivity [15]. If the primary purpose of these models is to distinguish the high performers from the rest (i.e., drafted versus not-drafted), sensitivity, or the correct classification of true positives should be considered with high priority. Artificial neural networks are becoming more widely used within sport, given their ability to identify trends in complex, real-world datasets that are non-linear in nature, where data are often not normally distributed [16]. With applications in prediction and classification tasks, neural networks operate in a manner that mimics the functionality of the human brain and its decision-making capacity by analysing the interaction of numerous complex variables [15]. For example, neural networks have been employed in soccer with performance variables from matches in the English Championship used to predict a players career trajectory by determining if they would be promoted to a higher league, continue in the current league, or play in lower leagues in future (78.8% success rate) [15].

The ability to use routinely collected elite-junior AF player data to predict draft success with high sensitivity would be useful to recruiters, clubs, coaches and players. Therefore, using logistic regression and neural networks, the aim of this study was to compare model performance when predicting player draft outcome during the 2021 AFL National Draft.

## Materials and methods

Physical testing, in-game movement (from Global Positioning Systems; GPS), and technical involvement data were collated from 708 elite-junior male Australian Rules football players

competing in the under 18 boys NAB League competition during five consecutive seasons (2017 to 2021). Data from the 2017 to 2020 seasons were made available at the conclusion (Sep 30[th]) of the 2020 competitive season. Data from the 2021 season were made available again at the conclusion of the competitive season allowing sufficient time to process prior to the National Draft on 24[th] November. Data were organised to only include those participants who were eligible (18[th] year) in each respective year. Data from eligible players (n = 465; drafted = 90, not-drafted = 375) who competed in the 2017 to 2020 seasons were used to construct logistic regression and neural network models with draft outcome (drafted or not-drafted) being the binary response variable. Player data from the 2021 season (n = 243) were run through the previously constructed models to prospectively predict draft success prior to the 2021 draft. Model performance was then analysed post. Access to archived data were granted by the Australian Football League. Institutional ethics approval with waived participant consent was granted by Latrobe University Human Ethics Committee (ref: HEC20065).

Physical testing outcomes were determined in March prior to the commencement of each season, and included: stature (cm), reach (cm), body mass (kg), vertical jump (cm), running vertical jump left (RVJL; cm), running vertical jump right (RVJR; cm), 20-m sprint (s), AFL Agility (s) and the Yo-Yo Intermittent Recovery Test (Estimated $\dot{V}O_2$ max; ml·kg$^{-1}$·min$^{-1}$). Estimated $\dot{V}O_2$ max was used to maintain consistency with the previous literature from which the logistic regression models were generated. Time (s) recorded in the 20-m sprint test was converted to an average speed (m·s$^{-1}$) and the time recorded to complete the AFL Agility test was subtracted from 10 s to ensure a faster time on both tests was represented by a larger number [4].

In-game movement data were collected at 10 Hz using GPS (Optimeye X4/S5; Catapult Innovations, Melbourne, Australia) as a standard procedure during 331 matches (5,240 appearances; mean = 13 ± 7 appearances per player). GPS variables assessed were software-derived and included field time (min), total distance (m), relative distance (total distance/field time [m·min$^{-1}$]), high speed running (HSR) efforts and sprint efforts. The velocity thresholds used for HSR and sprint efforts were 4.00 to 5.99 m·s$^{-1}$ and $\geq$6.00 m·s$^{-1}$, respectively.

Technical involvement data were collected by an external provider (Champion Data$^{TM}$, Melbourne, Australia) as a standard procedure during the same 331 matches and made available for analyses in their raw format with associated timestamps and player name. Variables were grouped for analyses to include relative involvements (n·min$^{-1}$), relative disposals (n·min$^{-1}$), relative possessions (n·min$^{-1}$), relative pressure acts (n·min$^{-1}$), and relative positive involvements (n·min$^{-1}$).

Players were assigned specific positions by coaches during physical testing. For analysis purposes, players were then assigned to an all-position group and three positional groups (nomadic, fixed, fixed&ruck). Due to their small sample (n = 15), ruckmen were combined with fixed-position players to form the fixed&ruck group. Ruckmen have comparable positional roles and physical attributes to fixed-position players [17]. Variables were collected from physical, GPS and technical data. To limit the impact of highly correlated variables and reduce the number of covariates, factor analysis using principal components analysis with oblique rotation was performed prior to logistic regression on all available variables (Version 26 IBM SPSS Statistics for Windows; IBM Corp, Armonk NY, USA). Underlying latent factors were identified using loading scores, which were then used as covariates in logistic regression models [4]. Variables that did not load on one specific factor were treated as their own covariate. For comparative purposes, logistic regression was also performed using unfactored data.

Evolutionary algorithms were used to define the architecture, optimisation methods, and parameters for the neural network models in a customised software (Analysis and

Recommendation Engine; PhysiGo Ltd., Wiltshire, England). The evolutionary component of the analysis process incorporated four parameters for building the network (number of layers, number of neurons in hidden layers, layer connectivity and their respective weights). The parameters that defined the training optimisation method were back propagation, Bayesian regulated optimisation and gaussian random optimisation. For back propagation parameters included learning rate, momentum terms, squashing terms, convergence terms, and a fitness value ($\chi^2$) to compare models. Parameters were encoded in a gene (eDNA). Retrospective data were randomly split into three subsets, as follows: a third is used to optimise the neural network given the parameters defined by the evolutionary algorithms, a third is used only to optimise the evolutionary algorithm based on the result of the neural network, a third to independently test the model on unseen data. This network design ensures no overfitting and avoids local minima and bias introduction. During model development, an eDNA string was selected from one of several gene pools (tribes) to build and optimise the neural network. It then used the eDNA to run the defined optimisation technique on a model and evaluate the eDNA's fitness. Fitness was then used to steer further model selection. Gene pools and tribes were set to compete against each other in a parallel modelling environment. Neural networks used the area under the curve of a Receiver Operator Characteristics curve to evaluate true/false positives/negatives in a confusion matrix.

For each respective model, players were ranked on their probability of being drafted in Microsoft Excel (Microsoft Corporation, Washington, USA). To evaluate the relative performance of models, cut-off thresholds ranging from 5% to 50% were used to predict 2021 draft outcome by player and data configuration in each positional group. Players ranked within the respective cut-off thresholds were allocated the binary outcome of drafted, and players ranked outside the cut-off thresholds were allocated the binary outcome of not-drafted. Following the draft, confusion matrices were constructed using the predicted and observed outcomes. Model performance was evaluated using specificity (number of true not-drafted/number of all not-drafted), sensitivity (number of true drafted/number of all drafted) and accuracy (number of correct assessments/number of all assessments).

## Results

Of the 243 players in the 2021 draft for which data were available, 38 (16%) were drafted and 205 (84%) were not-drafted. Fig 1 shows the distribution of drafted players per quartile by model and analysis type. In comparison with logistic regression, neural networks correctly predicted a greater proportion of drafted players in the first quartile (61 ± 6% *vs*. 41 ± 15%) and in the first two quartiles combined (85 ± 10% *vs*. 68 ± 17%). Factors identified from principal components analysis and used in modelling are presented in S1 Table. Listwise deletion was applied in logistic regression where cases had one or more missing data points (n = 22), which was not the case with neural networks. Consequently, prospective predictions were performed on all-position (logistic: n = 214; neural networks: n = 243), nomadic (logistic: n = 160; neural networks: n = 182,), fixed (logistic: n = 39; neural networks: n = 46), and fixed&ruck (logistic: n = 54; neural networks: n = 61).

Logistic regression and neural network performance comparisons (specificity, sensitivity) at progressive cut-off thresholds are presented in Fig 2. Table 1 presents the best performing models for each positional group (40 models) at each cut-off threshold used. Neural networks accounted for 29 (73%) of the 40 best performing models. When variables were factored, the difference in performance for logistic regression (specificity = 76 ± 13%; sensitivity = 49 ± 24%; accuracy = 72 ± 9%) and neural networks (specificity = 78 ± 13%; sensitivity = 55 ± 24%; accuracy = 74 ± 8%) across cut-off thresholds was small and overall performance at optimal

**Fig 1. Distribution of successful draft picks per quartile by model and analysis.**

thresholds were similar. Conversely, when variables were treated independently, performance of neural networks (specificity = 79 ± 13%; sensitivity = 61 ± 24%; accuracy = 76 ± 8%) was consistently superior to logistic regression (specificity = 73 ± 15%; sensitivity = 29 ± 14%; accuracy = 66 ± 11%).

Table 2 presents the model performance specifically at the 15% and 35% drafted cut-off thresholds. The selected cut-off thresholds of 15% and 35% were chosen because the proportion of observed drafted players per position group was ~15%, and the 35% cut-off threshold represents the most common first instance of convergence (i.e., intersection) of specificity and sensitivity, observed in 8 of 12 cases (Fig 1). In total, Table 2 presents 48 models with 24 comparisons. At the 15% and 35% cut-off thresholds, neural networks correctly classified a greater proportion of drafted players than logistic regression in 21 of the 24 comparisons. At the 15% cut-off threshold, neural networks (mean = 37.3%) had greater sensitivity than logistic

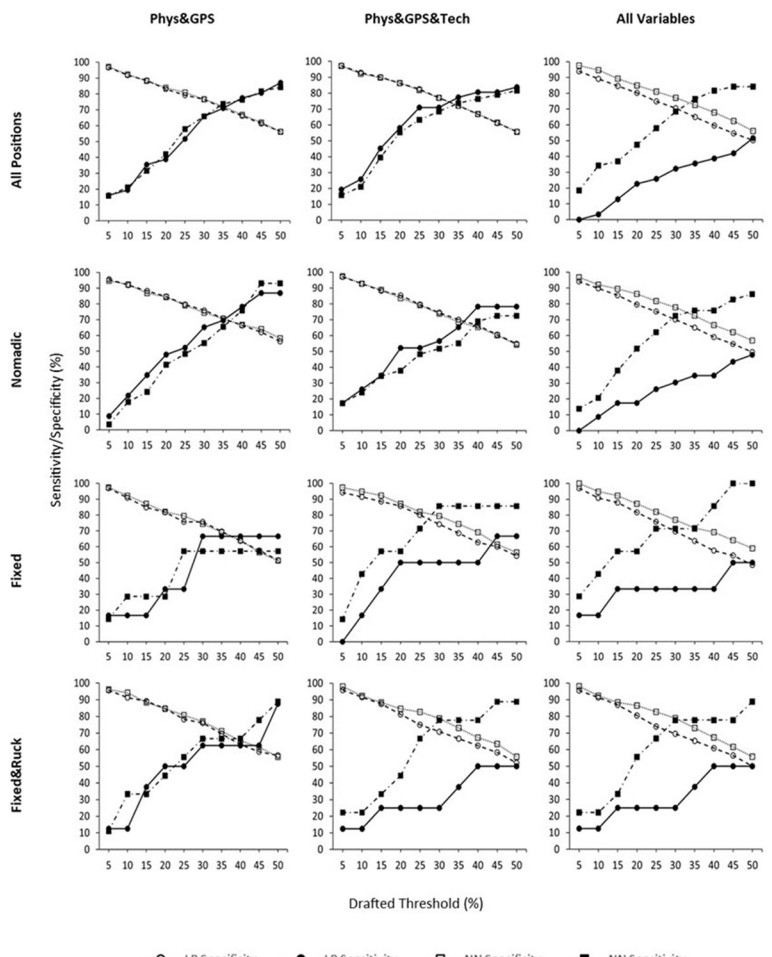

**Fig 2. Receiver Operator Characteristic curve comparisons of neural network (NN) and logistic regression (LR) models using relative sample cut-off thresholds from 5% to 50%.** LR–logistic regression, NN–neural networks.

regression models (mean = 29.3%). At the 35% cut-off threshold, neural networks (mean = 71.3%) had greater sensitivity than logistic regression models (mean = 53.4%).

## Discussion

This is the first study in Australian Rules football to systematically collect multifactorial, league-wide data over consecutive seasons and predict draft outcome. Thirty-eight players in which data were available were drafted. Logistic regression equations and neural networks were used, with resultant specificity, sensitivity and accuracy used to compare model performance. When reporting the best performing model for each positional group and data configuration, 73% of those were neural network models. Further, neural networks correctly classified a greater proportion of drafted players than logistic regression. However, when variables were factored, the difference in performance for logistic regression and neural networks across cut-off thresholds was trivial and overall performance at optimal thresholds were similar. Conversely, when data were not factored, performance of neural networks was consistently superior to logistic regression.

**Table 1. Best performing model specificity, sensitivity and accuracy for each position and cut-off threshold.**

| Threshold | Position | Model Type | Data | Specificity % (95% CI) | Sensitivity % (95% CI) | Accuracy % (95% CI) |
|---|---|---|---|---|---|---|
| 5% | All-Position | Logistic regression | Phys&GPS&Tech | 97.3 (93.7–99.1) | 19.4 (7.5–37.5) | 86.0 (80.6–90.3) |
| | Nomadic | Logistic regression | Phys&GPS&Tech | 97.1 (92.7–99.2) | 17.4 (5.0–38.8) | 85.6 (79.2–90.7) |
| | Fixed | Neural networks | All Variables | 100.0 (91.0–100.0) | 28.6 (3.7–71.0) | 89.1 (76.4–96.4) |
| | Fixed&Ruck | Neural networks | All Variables | 98.1 (89.7–100.0) | 22.2 (2.8–60.0) | 86.9 (75.8–94.2) |
| | | | Phys&GPS&Tech | | | |
| 10% | All-Position | Neural networks | All Variables | 94.6 (90.6–97.3) | 34.2 (19.6–51.4) | 85.2 (80.1–89.4) |
| | Nomadic | Logistic regression | Phys&GPS&Tech | 92.7 (87.0–96.4) | 26.1 (10.2–48.4) | 83.1 (76.4–88.6) |
| | Fixed | Neural networks | All Variables | 94.9 (82.7–99.4) | 42.9 (9.9–81.6) | 87.0 (73.7–95.1) |
| | | | Phys&GPS&Tech | | | |
| | Fixed&Ruck | Neural networks | Phys&GPS | 94.2 (84.1–98.8) | 33.3 (7.5–70.1) | 85.3 (73.8–93.0) |
| 15% | All-Position | Logistic regression | Phys&GPS&Tech | 90.2 (84.9–94.1) | 45.2 (27.3–64.0) | 83.6 (78.0–88.3) |
| | Nomadic | Neural networks | All Variables | 89.5 (83.6–93.9) | 37.9 (20.7–57.7) | 81.3 (74.9–86.7) |
| | Fixed | Neural networks | All Variables | 92.3 (79.1–98.4) | 57.1 (18.4–90.1) | 87.0 (73.7–95.1) |
| | | | Phys&GPS&Tech | | | |
| | Fixed&Ruck | Logistic regression | Phys&GPS | 89.1 (76.4–96.4) | 37.5 (8.5–75.5) | 81.5 (68.6–90.8) |
| 20% | All-Position | Logistic regression | Phys&GPS&Tech | 86.3 (80.5–91.0) | 58.1 (39.1–75.5) | 82.2 (76.5–87.1) |
| | Nomadic | Neural networks | All Variables | 86.3 (79.8–91.3) | 51.7 (32.5–70.6) | 80.8 (74.3–86.2) |
| | Fixed | Neural networks | All Variables | 87.2 (72.6–95.7) | 57.1 (18.4–90.1) | 82.6 (68.6–92.2) |
| | | | Phys&GPS&Tech | | | |
| | Fixed&Ruck | Neural networks | All Variables | 86.5 (74.2–94.4) | 55.6 (21.2–86.3) | 82.0 (70.0–90.6) |
| 25% | All-Position | Logistic regression | Phys&GPS&Tech | 82.5 (76.2–87.7) | 71.0 (52.0–85.8) | 80.8 (74.9–85.9) |
| | Nomadic | Neural networks | All Variables | 81.7 (74.7–87.5) | 62.1 (42.3–79.3) | 78.6 (71.9–84.3) |
| | Fixed | Neural networks | All Variables | 82.1 (66.5–92.5) | 71.4 (29.0–96.3) | 80.4 (66.1–90.6) |
| | | | Phys&GPS&Tech | | | |
| | Fixed&Ruck | Neural networks | All Variables | 82.7 (69.7–91.8) | 66.7 (29.9–92.5) | 80.3 (68.2–89.4) |
| 30% | All-Position | Logistic regression | Phys&GPS&Tech | 77.1 (70.3–82.9) | 71.0 (52.0–85.8) | 76.2 (69.9–81.7) |
| | Nomadic | Neural networks | All Variables | 77.8 (70.4–84.1) | 72.4 (52.8–87.3) | 76.9 (70.1–82.8) |
| | Fixed | Neural networks | Phys&GPS&Tech | 79.5 (63.5–90.7) | 85.7 (42.1–99.6) | 80.4 (66.1–90.6) |
| | Fixed&Ruck | Neural networks | All Variables | 78.9 (65.3–88.9) | 77.8 (40.0–97.2) | 78.7 (66.3–88.1) |
| | | | Phys&GPS&Tech | | | |
| 35% | All-Position | Neural networks | All Variable | 72.7 (66.0–78.7) | 76.3 (59.8–88.6) | 73.3 (67.2–78.7) |
| | Nomadic | Neural networks | All Variables | 72.6 (64.8–79.5) | 75.9 (56.5–89.7) | 73.1 (66.0–79.4) |
| | Fixed | Neural networks | Phys&GPS&Tech | 74.4 (57.9–87.0) | 85.7 (42.1–99.6) | 76.1 (61.2–87.4) |
| | Fixed&Ruck | Neural networks | All Variables | 73.1 (59.0–84.4) | 77.8 (40.0–97.2) | 73.8 (60.9–84.2) |
| | | | Phys&GPS&Tech | | | |
| 40% | All-Position | Neural networks | All Variables | 67.8 (60.9–74.1) | 81.6 (65.7–92.3) | 70.0 (63.8–75.7) |
| | Nomadic | Logistic regression | Phys&GPS | 66.4 (57.9–74.3) | 78.3 (56.3–92.5) | 68.1 (60.3–75.3) |
| | | | Phys&GPS&Tech | | | |
| | Fixed | Neural networks | All Variables | 69.2 (52.4–83.0) | 85.7 (42.1–99.6) | 71.7 (56.5–84.0) |
| | | | Phys&GPS&Tech | | | |
| | Fixed&Ruck | Neural networks | All Variables | 67.3 (52.9–79.7) | 77.8 (40.0–97.2) | 68.9 (55.7–80.1) |
| | | | Phys&GPS&Tech | | | |
| 45% | All-Position | Neural networks | All Variables | 62.4 (55.4–69.1) | 84.2 (68.8–94.0) | 65.8 (59.5–71.8) |
| | Nomadic | Neural networks | Phys&GPS | 64.1 (55.9–71.6) | 93.1 (77.2–99.2) | 68.7 (61.4–75.3) |
| | Fixed | Neural networks | All Variables | 64.1 (47.2–78.8) | 100.0 (59.0–100.0) | 69.6 (54.3–82.3) |
| | Fixed&Ruck | Neural networks | Phys&GPS&Tech | 63.5 (49.0–76.4) | 88.9 (51.8–99.7) | 67.2 (54.0–78.7) |

(*Continued*)

**Table 1.** (Continued)

| Threshold | Position | Model Type | Data | Specificity | Sensitivity | Accuracy |
|---|---|---|---|---|---|---|
| | | | | % (95% CI) | % (95% CI) | % (95% CI) |
| 50% | All-Position | Logistic regression | Phys&GPS | 56.3 (48.8–63.6) | 87.1 (70.2–96.4) | 60.8 (53.9–67.3) |
| | Nomadic | Neural networks | Phys&GPS | 58.2 (49.9–66.1) | 93.1 (77.2–99.2) | 63.7 (56.3–70.7) |
| | Fixed | Neural networks | All Variables | 59.0 (42.1–74.4) | 100.0 (59.0–100.0) | 65.2 (49.8–78.7) |
| | Fixed&Ruck | Logistic regression | Phys&GPS | 56.5 (41.1–71.1) | 87.5 (47.4–99.7) | 61.1 (46.9–74.1) |

Note: Phys- physical, GPS–global positioning system, Tech–technical, CI–Confidence Interval.

Neural networks were better than logistic regression at identifying drafted players in the first quartile and also in the first two quartiles combined (85 ± 10%). Comparative Receiver Operator Characteristics curves were used to compare model performance across a range of relative cut-off thresholds (5% to 50%). In 8 of the 12 model comparisons presented in Fig 1, the specificity and sensitivity curves intersected at lower cut-off thresholds in neural networks. Meaning, at any cut-off threshold before the point of intersection, neural networks correctly classified more true positive and true negative outcomes than logistic regression at the same cut-off threshold. In the same 8 comparisons, model accuracy (number of correct assessments/ number of all assessments) also remained higher than logistic regression after this point of intersection.

Neural network models had greater sensitivity than logistic regression, whereas logistic regression had exceptionally high specificity. Model sensitivity is important when the recruiters' aim is to identify players possessing characteristics that differentiate them from the majority of other players (i.e., draft potential). A model with high sensitivity is successfully classifying a high proportion of players that are actually drafted. Conversely, high specificity signifies a model that correctly classifies a high proportion of players that are not-drafted. As model accuracy reflects both sensitivity and specificity, and because not-drafted players (NAB League = 85%) significantly outweigh drafted players (NAB League = 15%), a model with high sensitivity and specificity is more desirable than a model with low sensitivity and very high specificity, where the overall accuracy is comparable.

These outcomes have implications in two important draft scenarios. Teams are allocated draft picks based on their ladder position the previous season, and any trades made internally between clubs. Approximately 120 players are drafted each year [18]. Clubs with early draft picks need to confidently (sensitivity) identify the talent that will immediately enhance their playing list. Smaller cut-off thresholds (e.g., 15%) with no positional bias would be useful to recruiters as neural networks in this instance are correctly classifying up to 15 drafted NAB League players. Conversely, as the draft progresses, clubs may opt for larger cut-off thresholds (e.g., 35%) or position-specific models that identify the talent to fill certain positional voids in their current playing list. At the 35% cut-off threshold, neural networks are correctly classifying 22/29 drafted nomadic players and 7/9 drafted fixed&ruck players.

The addition of a technical factor to logistic regression models resulted in improved model performance when comparing factored (Phys&GPS and Phys&GPS&Tech) models. When individual variables were used, performance was negatively influenced. This result is as expected given the limitations of logistic regression when dealing with multiple variables, many of which may be highly correlated (e.g., VJ and RVJL or RVJR) [19]. In contrast, draft prediction using neural networks was less effective using factored data and more effective when using individual variables. If modelling is undertaken for the purpose of training prescription, logistic regression using factored data provides insight on the impact of generalised

**Table 2. Model performance by position at the 15% and 35% drafted cut-off thresholds.**

| Position | Model Type | Data | N | Observed Drafted n (%) | Observed Not-Drafted n (%) | Correct Drafted (Sensitivity %) | Correct Not-Drafted (Specificity %) | Accuracy % (95% CI) |
|---|---|---|---|---|---|---|---|---|
| All-Position 15% | Logistic regression | Phys&GPS | 214 | 31 (14.5) | 183 (85.5) | 11 (35.5) | 162 (88.5) | 80.8 (74.9–85.9) |
| | | Phys&GPS&Tech | | | | 14 (45.2) | 165 (90.2) | 83.6 (78.0–88.3) |
| | | All Variables | | | | 4 (12.9) | 155 (84.7) | 74.3 (67.9–80.0) |
| | Neural networks | Phys&GPS | 243 | 38 (15.6) | 205 (84.4) | 12 (31.6) | 181 (88.3) | 79.4 (73.8–84.3) |
| | | Phys&GPS&Tech | | | | 15 (39.5) | 184 (89.8) | 81.9 (76.5–86.5) |
| | | All Variables | | | | 14 (36.8) | 183 (89.3) | 81.1 (75.6–85.8) |
| Nomadic 15% | Logistic regression | Phys&GPS | 160 | 23 (14.4) | 137 (85.6) | 8 (34.8) | 121 (88.3) | 80.6 (73.6–86.4) |
| | | Phys&GPS&Tech | | | | 8 (34.8) | 121 (88.3) | 80.6 (73.6–86.4) |
| | | All Variables | | | | 4 (17.4) | 117 (85.4) | 75.6 (68.2–82.1) |
| | Neural networks | Phys&GPS | 182 | 29 (15.9) | 153 (84.1) | 7 (24.1) | 133 (86.9) | 76.9 (70.1–82.8) |
| | | Phys&GPS&Tech | | | | 10 (34.5) | 136 (88.9) | 80.2 (73.7–85.7) |
| | | All Variables | | | | 11 (37.9) | 137 (89.5) | 81.3 (74.9–86.7) |
| Fixed 15% | Logistic regression | Phys&GPS | 39 | 6 (15.4) | 33 (84.6) | 1 (16.7) | 28 (84.8) | 74.4 (57.9–87.0) |
| | | Phys&GPS&Tech | | | | 2 (33.3) | 31 (93.9) | 80.5 (65.1–91.2) |
| | | All Variables | | | | 2 (33.3) | 29 (87.9) | 79.5 (63.5–90.7) |
| | Neural networks | Phys&GPS | 46 | 7 (15.2) | 39 (84.8) | 2 (28.6) | 34 (87.2) | 78.3 (63.6–89.1) |
| | | Phys&GPS&Tech | | | | 4 (57.1) | 36 (92.3) | 87.0 (73.7–95.1) |
| | | All Variables | | | | 4 (57.1) | 36 (92.3) | 87.0 (73.7–95.1) |
| Fixed&Ruck 15% | Logistic regression | Phys&GPS | 54 | 8 (14.8) | 46 (85.2) | 3 (37.5) | 41 (89.1) | 81.5 (68.6–90.8) |
| | | Phys&GPS&Tech | | | | 2 (25.0) | 40 (87.0) | 78.6 (65.6–88.4) |
| | | All Variables | | | | 2 (25.0) | 40 (87.0) | 77.8 (64.4–88.0) |
| | Neural networks | Phys&GPS | 61 | 9 (14.8) | 52 (85.2) | 3 (33.3) | 46 (88.5) | 80.3 (68.2–89.4) |
| | | Phys&GPS&Tech | | | | 3 (33.3) | 46 (88.5) | 80.3 (68.2–89.4) |
| | | All Variables | | | | 3 (33.3) | 46 (88.5) | 80.3 (68.2–89.4) |
| All-Position 35% | Logistic regression | Phys&GPS | 214 | 31 (14.5) | 183 (85.5) | 22 (71.0) | 130 (71.0) | 71.0 (64.5–77.0) |
| | | Phys&GPS&Tech | | | | 24 (77.4) | 132 (72.1) | 72.9 (66.4–78.7) |
| | | All Variables | | | | 11 (35.5) | 119 (65.0) | 60.8 (53.9–67.3) |
| | Neural networks | Phys&GPS | 243 | 38 (15.6) | 205 (84.4) | 28 (73.7) | 148 (72.2) | 72.4 (66.4–78.0) |
| | | Phys&GPS&Tech | | | | 28 (73.7) | 148 (72.2) | 72.4 (66.4–78.0) |
| | | All Variables | | | | 29 (76.3) | 149 (72.7) | 73.3 (67.2–78.7) |
| Nomadic 35% | Logistic regression | Phys&GPS | 160 | 23 (14.4) | 137 (85.6) | 16 (69.6) | 97 (70.8) | 70.6 (62.9–77.6) |
| | | Phys&GPS&Tech | | | | 15 (65.2) | 96 (70.1) | 69.4 (61.6–76.4) |
| | | All Variables | | | | 8 (34.8) | 89 (65.0) | 60.6 (52.6–68.3) |
| | Neural networks | Phys&GPS | 182 | 29 (15.9) | 153 (84.1) | 19 (65.5) | 108 (70.6) | 69.8 (62.6–76.4) |
| | | Phys&GPS&Tech | | | | 16 (55.2) | 105 (68.6) | 66.5 (59.1–73.3) |
| | | All Variables | | | | 22 (75.0) | 111 (72.6) | 73.1 (66.0–79.4) |
| Fixed 35% | Logistic regression | Phys&GPS | 39 | 6 (15.4) | 33 (84.6) | 4 (66.7) | 23 (69.7) | 69.2 (52.4–83.0) |
| | | Phys&GPS&Tech | | | | 3 (50.0) | 24 (68.6) | 65.9 (49.4–79.9) |
| | | All Variables | | | | 2 (33.3) | 21 (63.6) | 59.0 (42.1–74.4) |
| | Neural networks | Phys&GPS | 46 | 7 (15.2) | 39 (84.8) | 4 (57.1) | 27 (69.2) | 67.4 (52.0–80.5) |
| | | Phys&GPS&Tech | | | | 6 (85.7) | 29 (74.4) | 76.1 (61.2–87.4) |
| | | All Variables | | | | 5 (71.4) | 28 (71.8) | 71.7 (56.5–84.0) |

(*Continued*)

**Table 2.** (Continued)

| Position | Model Type | Data | N | Observed Drafted | Observed Not-Drafted | Correct Drafted (Sensitivity %) | Correct Not-Drafted (Specificity %) | Accuracy % (95% CI) |
|---|---|---|---|---|---|---|---|---|
| | | | | n (%) | n (%) | | | |
| Fixed&Ruck 35% | Logistic regression | Phys&GPS | 54 | 8 (14.8) | 46 (85.2) | 5 (62.5) | 32 (69.6) | 68.5 (54.5–80.5) |
| | | Phys&GPS&Tech | | | | 3 (37.5) | 32 (66.7) | 62.5 (48.6–75.1) |
| | | All Variables | | | | 3 (37.5) | 30 (65.2) | 61.1 (46.9–74.1) |
| | Neural networks | Phys&GPS | 61 | 9 (14.8) | 52 (85.2) | 6 (66.7) | 37 (71.2) | 70.5 (57.4–81.5) |
| | | Phys&GPS&Tech | | | | 7 (77.8) | 38 (73.1) | 73.8 (60.9–84.2) |
| | | All Variables | | | | 7 (77.8) | 38 (73.1) | 73.8 (60.9–84.2) |

Note: N—number of players, Phys—physical, GPS–global positioning system, Tech–technical.

capacities that influence draft success. However, specific domain knowledge is required to determine the number and relatability of the selected input factors. Conversely, neural networks can include more variables across a broad range of factors (e.g., S1 Table) and offer a more streamlined workflow by removing the need to perform factorisation.

There are a number of limitations that must be acknowledged within this study. Firstly, while positional analyses were used for exploratory purposes in this study, classifying players into positional groups can be problematic given that individuals can play multiple positions throughout the season, especially in elite-junior competition. Positional outcomes from this study should therefore be viewed with caution. Second, it is acknowledged that the data used in this study is solely from Victorian based teams. While it is assumed that data from other state-based talent competitions would be similar when making recommendations, this is not supported by findings. Third, when inspecting the data, the 95% CI of neural networks are somewhat larger than logistic regression. While neural network mean sensitivity values in particular are generally higher, the CI's often overlap logistic regression results, meaning conclusions must be taken with caution. Fourth, the models presented have been developed using drafted and not-drafted data outcome data from previous seasons. Consequently, any decision to recruit a player based on outcomes from these models indicates a similarity to the type of player that has been drafted in the past and does not consider the success of the player after being drafted. As more data becomes available, future modelling could use a measurement of career success as the dependent variable. Finally, the absence of psychosocial variables within these analyses must be acknowledged. Researchers have identified the importance of certain psychological characteristics that can impact a career in elite sport. Characteristics deemed important include self-confidence, drive or motivation, commitment, mental toughness and resilience [1, 18]. It is now common practice to assess these characteristics through interviewing procedures within the AFL draft and if made available, could be considered within future analyses [3].

Collectively, the findings from this study provide justification for the application of predictive modelling for both retrospective and prospective classification tasks like the AFL draft. If domain knowledge exists, the recommendation would be to use all available variables and neural networks for model development and prediction. If not, factored data and logistic regression offers an alternative solution, but these models have greater variability in predicting drafted players across positions and models. Recruiters could use data from multiple seasons to iteratively train neural network models and periodically predict draft outcome to add weight to their subjective selections. The process of predicting draft likelihood with this method might additionally identify players that had not previously been considered, or highlight trends

in player performance that could aid or hinder their transition into the elite-senior game. For example, an upward trending athlete that did not feature in mid-season predictions, but did feature in post-season predictions, might be a more desirable draft pick than a downward trending player. Talent pathway coaches could use this information to objectively identify players with "current" draft potential and ensure they are developed to continue on that trajectory. Alternatively, identifying players who are ranked close to the cut-off threshold, and variables that they are deficient in, could inform targeted development for an individual player to improve their chances of being drafted.

## Conclusion

This is the first study to systematically collect multifactorial, league-wide data over consecutive seasons and prospectively predict draft outcome in elite-junior Australian Rules football. Where the goal of analysis is to identify talented players with draft potential, model sensitivity is paramount, and neural networks generally outperformed logistic regression. Employing neural networks removes the need to factor data and resulted in superior classification of talented individuals when compared to logistic regression models. If logistic regression is to be used, data should be factored, and results applied with more caution. Future research could further develop these predictive models by including psychosocial characteristics, and players from other elite-junior talent pathway competitions.

## Supporting information

**S1 Table. Factor variables.**
(DOCX)

## Acknowledgments

This work was supported by an Australian Government Research Training Program Scholarship with an industry focus through the involvement of AFL Victoria. We also acknowledge the support of the Bendigo Tertiary Education Anniversary Foundation and Holsworth Research Initiative for Professor Kingsley's research.

## Author Contributions

**Conceptualization:** Jacob Jennings, Daniel W. Wundersitz, Courtney J. Sullivan, Stephen D. Cousins, Michael I. Kingsley.

**Data curation:** Jacob Jennings, Jay C. Perrett.

**Formal analysis:** Jacob Jennings, Jay C. Perrett.

**Methodology:** Jacob Jennings, Michael I. Kingsley.

**Writing – original draft:** Jacob Jennings.

**Writing – review & editing:** Jacob Jennings, Daniel W. Wundersitz, Courtney J. Sullivan, Stephen D. Cousins, Michael I. Kingsley.

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
