## [Decision Letter · Decision Letter 0]

15 Dec 2023

PONE-D-23-37704Predicting successful draft outcome in Australian Rules football: model sensitivity is superior in neural networks when compared to logistic regression.PLOS ONE

Dear Dr. Kingsley,

Thank you for submitting your manuscript to PLOS ONE. After careful consideration, we feel that it has merit but does not fully meet PLOS ONE’s publication criteria as it currently stands. Therefore, we invite you to submit a revised version of the manuscript that addresses the points raised during the review process.

We look forward to receiving your revised manuscript.

Kind regards,

Julio Alejandro Henriques Castro da Costa

Academic Editor

PLOS ONE

Journal Requirements:

Reviewers' comments:

Reviewer's Responses to Questions

**Comments to the Author**

1. Is the manuscript technically sound, and do the data support the conclusions?

Reviewer #1: Yes

Reviewer #2: Yes

2. Has the statistical analysis been performed appropriately and rigorously? 

Reviewer #1: Yes

Reviewer #2: Yes

3. Have the authors made all data underlying the findings in their manuscript fully available?

Reviewer #1: Yes

Reviewer #2: No

4. Is the manuscript presented in an intelligible fashion and written in standard English?

Reviewer #1: Yes

Reviewer #2: Yes

5. Review Comments to the Author

Reviewer #1: This paper explores an interesting topic and is well written. The use of neural networks in this area is novel and informative. The methods are thorough and clearly explained. I think some tempering of the results is needed given that the confidence intervals for accuracy of LR and NN models almost always overlap; hence, the superiority of the NN approach is not certain. Additionally, the improved performance of the NN approach seems to be limited to using non-factored data. I have provided a small number of specific comments for the authors to consider below.

Introduction: The rationale for the study concept is clear but the introduction would benefit from greater recognition of the importance of predicting career success rather than simply getting drafted. It would seem more critical to predict who will experience career success post-draft rather than just identify who will get drafted. Clubs are more likely to benefit from knowing which player selections were effective vs not, rather than developing a model that is calibrated to make the same draft decisions they would have made anyway. If the authors disagree with this conceptual limitation, then the introduction would benefit from greater justification of why predicting drafted vs not drafted status will help clubs get greater benefit from their draft investments.

Methods: Please add details about the software used to develop and apply these analyses and neural networks. Those details appear to be missing from the methods.

Results:

Line #159: Can you please report on the distribution of draft pick numbers for the players who were selected in the draft? If these are clustered towards the end of the draft then your model may not be calibrated towards the highest performing players (i.e., those most likely to have career success).

Line #182: The confidence intervals for accuracy of LR and NN models almost always overlap, which implies uncertainty about superior performance. Please acknowledge this in your description of the Table 2 data.

Discussion

Line #231-238: Can you provide greater justification for why NN with unfactored data is advantageous over factored LR data analysis? It would seem that data analysis pipelines could be configured to support either approach, so I believe a deeper and more insightful discussion of which is practically more valuable is required.

Line #239: As per my comment for the introduction, I think there are limitations of using drafted vs non-drafted status as the outcome given that this does not consider draft position nor the reality that many draft selections prove to be ineffective and some undrafted players ends up breaking through to have quality careers through other progression pathways. This has impacts on the practical applications suggested in the following paragraph (starting at Line #254).

Line #257-259: However, sensitivity was similar between NN and LR for factored data so the basis for this directive is not clear from the reported results (i.e., 49 +/- 24% vs 55 +/- 24% sensitivity).

Line #277-78: as per above comment, it’s hard to buy the “low sensitivity” angle here when the NN and LR approaches were so close to equivalent for factored data (49 +/- 24% vs 55 +/- 24%) i.e., it seems strange to label 49% as low and not suggest the same for 55% when the practical implications of 49% and 55% sensitivity are so similar.

Reviewer #2: Firstly, I would like to thank the Authors for the opportunity to review this paper. It was an interesting read. This paper aimed to highlight the differences between logistic regression and neural network models in their application of predicting the AFL draft. The application of machine learning algorithms and non-linear approaches to estimating sport outcomes is extremely relevant and important to the sport landscape. Overall the study is well constructed and the paper well written. I only have some minor comments and questions before recommending this paper for publication.

Abstract

No comments

Intro

Line 74-77: This sentence is long and worded. If you could call out sensitivity and define it more clearly for the reader in a separate sentence this would be beneficial as it’s a key tenet of the paper.

Methods

Please include what software and/or packages were used to perform the analyses.

Line 115: Was the distribution of games recorded on GPS similar across all players in the sample? i.e. players who experience injury, deselection or device failure may have fewer recorded match outputs than other players.

Line 123: Please include details on who assigned player positions? Was this done by the statistics provider, coaches or the researchers?

Line 125: Do you mean here that you assigned other players, such as tall forwards or tall backs, to the ruckmen group to increase the group size? How did you determine if the group size was sufficient after adding the talls? Perhaps make this a bit clearer.

Line 128: Could you explain why you performed PCA prior to the logistic regression?

Results

Line 159: I think it could be helpful to state the % of drafter players in the 2021 season. And also prior in the test data. Just for the reader to easily see the % of classes in both the training and testing data.

Discussion

Line 203-204: Can you outline any thoughts on why do you think the factored data was much more beneficial for logistic regression? Just briefly.

Line 218-220: Great to call out the importance of sensitivity here – as mentioned this is important with unbalanced classes such as this.

Good applications to the AFL draft.

Line 234: Can you provide a reference for the limitations of logistic regression using correlated variables?

Could you discuss/ the limitation in application of identifying talent using only the value of draft vs not draft? Ultimately, the decision to draft a player will, and should, be based on the player’s potential for success in performance at AFL level, not just the success of being drafted. In many ways, it is a failure of the recruitment team to draft a player who goes on to perform poorly at AFL level. In application, basing draft predictions only on factors associated with historical draftees should be used in caution. i.e. a recruitment team who uses this model to make draft selections, will simply be selecting the types of players who have been drafted in previous years.

Line 263-269: No questions I just really like the applications you’ve stated here.

6. PLOS authors have the option to publish the peer review history of their article (what does this mean?). If published, this will include your full peer review and any attached files.

Reviewer #1: No

Reviewer #2: No

---

## [Author Response · Author response to Decision Letter 0]

20 Jan 2024

Dear reviewers and editors,

We would like to take the opportunity to thank both reviewers for reviewing this manuscript and the editorial board for the opportunity to revise. We believe that our revisions in response to your comments have improved the manuscript. Your contributions are greatly appreciated. 

Our point-by-point responses and actions to your comments are below. Modifications are presented in red text in the “marked-up” copy of the manuscript.

Reviewer #1: This paper explores an interesting topic and is well written. The use of neural networks in this area is novel and informative. The methods are thorough and clearly explained. I think some tempering of the results is needed given that the confidence intervals for accuracy of LR and NN models almost always overlap; hence, the superiority of the NN approach is not certain. Additionally, the improved performance of the NN approach seems to be limited to using non-factored data. I have provided a small number of specific comments for the authors to consider below.

Response: Thank you for your positive and constructive comments. Details of the revisions made in response to the specific comments are provided below.

Comment 1.1: Introduction - The rationale for the study concept is clear but the introduction would benefit from greater recognition of the importance of predicting career success rather than simply getting drafted. It would seem more critical to predict who will experience career success post-draft rather than just identify who will get drafted. Clubs are more likely to benefit from knowing which player selections were effective vs not, rather than developing a model that is calibrated to make the same draft decisions they would have made anyway. If the authors disagree with this conceptual limitation, then the introduction would benefit from greater justification of why predicting drafted vs not drafted status will help clubs get greater benefit from their draft investments.

Response 1.1: We acknowledge the importance of predicting career success following draft for teams in the AFL. However, the focus of this work was to inform the talent pathway, where draft outcome is considered to be the direct outcome. With this in-mind, the main application of this research within the AFL is to provide objective, data driven justification to support draft decision making, and to inform player development and training for recruited players. Nevertheless, and in light of your comments, we have taken the following specific actions in the introduction to clarify the purpose of this work and to acknowledge the potential importance of career success as a different outcome to investigate in future studies. 

Action 1.1: 

Lines 44-47 – deleted.

Lines 47-48 - “Additionally, identifying strengths and weaknesses in the characteristics of individual players that have potential to impact draft outcome can inform player development and training decisions.” Has been changed to “Identifying the strengths and weaknesses of individual players that impact draft success can inform recruitment, player development and training decisions.”

Lines 49-57 – This was a large paragraph that has now been split with some additional alterations. Paragraph two now reads “Within Australian Rules football (AF), recruitment occurs informally through scouting of the junior participation pathway into the elite-junior talent pathway and formally through a yearly structured draft system in the elite-senior Australian Football League (AFL). Similar to other draft-based sports like Basketball8 and American Football,9 talented juniors and off-contract players are invited to be part of the AFL’s National Draft in November of each year. Therefore, the primary objective of the elite-junior talent pathway is to provide AFL teams with new recruits who possess desirable attributes. Although being drafted does not guarantee a successful career in the AFL, draft success is a critical step in the talent pathway. Consequently, the potential exists to use data from the elite-junior talent pathway to reduce subjectivity in the selection process.5-7”

Lines 57-58 – deleted.

Comment 1.2: Methods - Please add details about the software used to develop and apply these analyses and neural networks. Those details appear to be missing from the methods.

Response 1.2: Thank you for highlighting this omission. We have adjusted the manuscript to include this information.

Action 1.2:

Lines 138–139 – Addition of “in a customised software (Analysis and Recommendation Engine; PhysiGo Ltd., Wiltshire, England).”

Comment 1.3: - Line #159: Can you please report on the distribution of draft pick numbers for the players who were selected in the draft? If these are clustered towards the end of the draft then your model may not be calibrated towards the highest performing players (i.e., those most likely to have career success).

Response 1.3: Thank you for this insightful comment. We have included a new figure to display the performance of these models by quartile of outcome rank. 

Action 1.3:

Addition of Figure 1 to the manuscript.

Lines 167-170 – Addition of “Figure 1 shows the distribution of drafted players per quartile by model and analysis type. In comparison with logistic regression, neural networks correctly predicted a greater proportion of drafted players in the first quartile (61 ± 6% vs. 41 ± 15%) and in the first two quartiles combined (85 ± 10% vs. 68 ± 17%).”

Lines 216-217 – Addition of “Neural networks were better than logistic regression at identifying drafted players in the first quartile and also in the first two quartiles combined (85 ± 10%).”

Comment 1.4: - Line #182: The confidence intervals for accuracy of LR and NN models almost always overlap, which implies uncertainty about superior performance. Please acknowledge this in your description of the Table 2 data.

Response 1.4: Thank you for your recommendation. 

Action 1.4:

Lines 262-264 – Addition of “While neural network mean sensitivity values in particular are generally higher, the CI’s often overlap logistic regression results, meaning conclusions must be taken with caution.” 

Comment 1.5: - Line #231-238: Can you provide greater justification for why NN with unfactored data is advantageous over factored LR data analysis? It would seem that data analysis pipelines could be configured to support either approach, so I believe a deeper and more insightful discussion of which is practically more valuable is required.

Response 1.5: Thank you for this suggestion. We have made alterations to the paragraph in question to more appropriately discuss the application of both types of analyses.

Action 1.5:

Lines 249-254 – “Neural networks offer a preferred method for streamlined draft prediction or talent identification, removing the need to factor data, however it is acknowledged that there is specific domain knowledge required for these analyses.” Was changed to “If modelling is undertaken for the purpose of training prescription, logistic regression using factored data provides insight on the impact of generalised capacities that influence draft success. However, specific domain knowledge is required to determine the number and relatability of the selected input factors. Conversely, neural networks can include more variables across a broad range of factors (e.g., Supplementary Table 1) and offer a more streamlined workflow by removing the need to perform factorisation.”

Comment 1.6: - Line #239: As per my comment for the introduction, I think there are limitations of using drafted vs non-drafted status as the outcome given that this does not consider draft position nor the reality that many draft selections prove to be ineffective and some undrafted players ends up breaking through to have quality careers through other progression pathways. This has impacts on the practical applications suggested in the following paragraph (starting at Line #254).

Line #257-259: However, sensitivity was similar between NN and LR for factored data so the basis for this directive is not clear from the reported results (i.e., 49 +/- 24% vs 55 +/- 24% sensitivity).

Line #277-78: as per above comment, it’s hard to buy the “low sensitivity” angle here when the NN and LR approaches were so close to equivalent for factored data (49 +/- 24% vs 55 +/- 24%) i.e., it seems strange to label 49% as low and not suggest the same for 55% when the practical implications of 49% and 55% sensitivity are so similar.

Response 1.6: Thank you for pointing these out. This fits into the whole theme of tempering the manuscript given these limitations. Changes have been made to rectify these statements which have been influenced by your previous recommendation to discuss the distribution of drafted picks within models.

Action 1.6:

Addition of Figure 1 to the manuscript.

Lines 278-280 – “If not, factored data and logistic regression would suffice, but understand that these models are likely going to be less effective at identifying the drafted players, and more effective at identifying the not-drafted players.” Was changed to “If not, factored data and logistic regression offers an alternative solution, but these models have greater variability in predicting drafted players across positions and models.”

Line 299 – removed “due to low sensitivity”.

Reviewer #2: Firstly, I would like to thank the Authors for the opportunity to review this paper. It was an interesting read. This paper aimed to highlight the differences between logistic regression and neural network models in their application of predicting the AFL draft. The application of machine learning algorithms and non-linear approaches to estimating sport outcomes is extremely relevant and important to the sport landscape. Overall the study is well constructed and the paper well written. I only have some minor comments and questions before recommending this paper for publication.

Response: Thank you for your positive feedback and contribution to the manuscript. We have considered all comments accordingly.

Comment 2.1: Line 74-77: This sentence is long and worded. If you could call out sensitivity and define it more clearly for the reader in a separate sentence this would be beneficial as it’s a key tenet of the paper.

Response 2.1: Thanks for picking this up. Adjustments have been made so that this is more clearly explained.

Action 2.1: Lines 75-78 - “If the primary purpose of these models is to identify players that possess talent-defining characteristics that distinguish them from the rest (i.e., drafted versus not-drafted), sensitivity is the correct classification of true positives, in this case a player being drafted, and should be considered as a measure with high priority.” Was changed to “If the primary purpose of these models is to distinguish the high performers from the rest (i.e., drafted versus not-drafted), sensitivity, or the correct classification of true positives should be considered with high priority.”

Comment 2.2: Please include what software and/or packages were used to perform the analyses.

Response 2.2: Thank you for this pick up. This was an oversight on our end.

Action 2.2:

Line 133 – Added “(Version 26 IBM SPSS Statistics for Windows; IBM Corp, Armonk NY, USA).”

Line 138 – Added “in a customised software (Analysis and Recommendation Engine; PhysiGo Ltd., Wiltshire, England).”

Line 155 – Added “Microsoft Excel (Microsoft Corporation, Washington, USA).”

Comment 2.3: Line 115: Was the distribution of games recorded on GPS similar across all players in the sample? i.e. players who experience injury, deselection or device failure may have fewer recorded match outputs than other players.

Response 2.3: Thank you for suggesting this addition. 

Action 2.3: Line 116 – Addition of “(5,240 appearances; mean = 13 ± 7 appearances per player).”

Comment 2.4: Line 123: Please include details on who assigned player positions? Was this done by the statistics provider, coaches or the researchers?

Line 125: Do you mean here that you assigned other players, such as tall forwards or tall backs, to the ruckmen group to increase the group size? How did you determine if the group size was sufficient after adding the talls? Perhaps make this a bit clearer.

Response 2.4: We have provided additional information regarding positional identification.

Action 2.4: Lines 126-130: “Players were assigned to an all-position group and one of three positional groups (nomadic, fixed or ruckmen). The small sample of ruckmen required them to be combined with fixed-position players who have comparable positional roles and physical attributes.” Was changed to “Players were assigned specific positions by coaches during physical testing. For analysis purposes, players were then assigned to an all-position group and three positional groups (nomadic, fixed, fixed&ruck). Due to their small sample (n=15), ruckmen were combined with fixed-position players to form the fixed&ruck group. Ruckmen have comparable positional roles and physical attributes to fixed-position players.”

Comment 2.5: Line 128: Could you explain why you performed PCA prior to the logistic regression?

Response 2.5: Thank you, this was an oversight on our behalf and we agree that this is required.

Action 2.5: Lines 130-131 – Addition of “To limit the impact of highly correlated variables and reduce the number of covariates,”

Comment 2.6: Line 159: I think it could be helpful to state the % of drafter players in the 2021 season. And also prior in the test data. Just for the reader to easily see the % of classes in both the training and testing data.

Response 2.6: Thank you for this suggestion. We have made the necessary adjustments in the results section. We would also like to point out that the number of drafted vs not drafted from the test data set is reported in absolute terms on line 106.

Action 2.6: Lines 166-167 altered to include (16%) and (84%).

Comment 2.7: Line 203-204: Can you outline any thoughts on why do you think the factored data was much more beneficial for logistic regression? Just briefly.

Response 2.7: Having considered this point, we are not sure the first overview paragraph of the discussion is the best place for this. However, we have made additions to the methods section (Lines 130-131) outlining why the factor analysis was performed. Additionally, lines 246-248 explain the limitations of logistic regression more specifically.

Comment 2.8: Line 218-220: Great to call out the importance of sensitivity here – as mentioned this is important with unbalanced classes such as this.

Good applications to the AFL draft.

Response 2.8: Thank you.

Comment 2.9: Line 234: Can you provide a reference for the limitations of logistic regression using correlated variables?

Response 2.9: This addition has been made. Thank you.

Action 2.9: Addition of reference 19 on line 248.

Comment 2.10: Could you discuss/ the limitation in application of identifying talent using only the value of draft vs not draft? Ultimately, the decision to draft a player will, and should, be based on the player’s potential for success in performance at AFL level, not just the success of being drafted. In many ways, it is a failure of the recruitment team to draft a player who goes on to perform poorly at AFL level. In application, basing draft predictions only on factors associated with historical draftees should be used in caution. i.e. a recruitment team who uses this model to make draft selections, will simply be selecting the types of players who have been drafted in previous years.

Response 2.10: Thank you for this suggestion. We agree that this is a limitation of the current study and have acknowledged this.

Action 2.10: Lines 264-269 - Added “Fourth, the models presented have been developed using drafted and not-drafted data outcome data from previous seasons. Consequently, any decision to recruit a player based on outcomes from these models indicates a similarity to the type of player that has been drafted in the past and does not consider the success of the player after being drafted. As more data becomes available, future modelling could use a measurement of career success as the dependent variable.”

Comment 2.11: Line 263-269: No questions I just really like the applications you’ve stated here.

Response 2.11: Thank you.

---

## [Decision Letter · Decision Letter 1]

30 Jan 2024

Predicting successful draft outcome in Australian Rules football: model sensitivity is superior in neural networks when compared to logistic regression.

PONE-D-23-37704R1

Dear Dr. Kingsley,

We’re pleased to inform you that your manuscript has been judged scientifically suitable for publication and will be formally accepted for publication once it meets all outstanding technical requirements.

Kind regards,

Julio Alejandro Henriques Castro da Costa

Academic Editor

PLOS ONE

Additional Editor Comments (optional):

Reviewers' comments:

Reviewer's Responses to Questions

**Comments to the Author**

1. If the authors have adequately addressed your comments raised in a previous round of review and you feel that this manuscript is now acceptable for publication, you may indicate that here to bypass the “Comments to the Author” section, enter your conflict of interest statement in the “Confidential to Editor” section, and submit your "Accept" recommendation.

Reviewer #1: All comments have been addressed

Reviewer #2: All comments have been addressed

2. Is the manuscript technically sound, and do the data support the conclusions?

Reviewer #1: (No Response)

Reviewer #2: Yes

3. Has the statistical analysis been performed appropriately and rigorously? 

Reviewer #1: (No Response)

Reviewer #2: Yes

4. Have the authors made all data underlying the findings in their manuscript fully available?

Reviewer #1: (No Response)

Reviewer #2: Yes

5. Is the manuscript presented in an intelligible fashion and written in standard English?

Reviewer #1: (No Response)

Reviewer #2: Yes

6. Review Comments to the Author

Reviewer #1: (No Response)

Reviewer #2: Thank you for taking the time to address the comments for this paper. I think this manuscript will make a great addition to this journal. I'm happy to submit a recommendation to accept.

Regards,

7. PLOS authors have the option to publish the peer review history of their article (what does this mean?). If published, this will include your full peer review and any attached files.

Reviewer #1: No

Reviewer #2: **Yes: **Ben Teune

---

## [Editor Report · Acceptance letter]

17 Feb 2024

PONE-D-23-37704R1 

PLOS ONE

Dear Dr. Kingsley, 

I'm pleased to inform you that your manuscript has been deemed suitable for publication in PLOS ONE. Congratulations! Your manuscript is now being handed over to our production team.

Kind regards, 

on behalf of

Dr. Julio Alejandro Henriques Castro da Costa 

Academic Editor

PLOS ONE